# Selenium Alleviates Cadmium Toxicity by Regulating Carbon Metabolism, AsA-GSH Cycle, and Cadmium Transport in *Glycyrrhiza uralensis* Fisch. Seedlings

**DOI:** 10.3390/plants14121736

**Published:** 2025-06-06

**Authors:** Xuerong Zheng, Jiafen Luo, Xin Li, Chaoyue Zhang, Guigui Wan, Caixia Xia, Jiahui Lu

**Affiliations:** 1College of Life Sciences, Shihezi University, Shihezi 832003, China; zxr11170526@163.com (X.Z.); x2492849969@163.com (J.L.); m17716952513@163.com (X.L.); 18899615812@163.com (C.Z.); 13150400690@163.com (G.W.); xcx17590936372@163.com (C.X.); 2Licorice Research Institute, Shihezi University, Shihezi 832003, China; 3Xinjiang Production and Construction Corps Key Laboratory of Oasis Town and Mountain-Basin System Ecology, Shihezi University, Shihezi 832003, China; 4Key Laboratory of Xinjiang Phytomedicine Resource and Utilization, Ministry of Education, Shihezi University, Shihezi 832003, China

**Keywords:** *Glycyrrhiza*, selenium, cadmium, physiology, transcriptome analysis

## Abstract

Cadmium (Cd) accumulation in plants hinders their growth and development while posing significant risks to human health through food chain transmission. *Glycyrrhiza uralensis* Fisch. (*G. uralensis*) is a medicinal plant valued for its roots and plays a crucial role in harmonizing various herbs in traditional Chinese medicine prescriptions. However, widespread Cd contamination in soil limits safe cultivation and application. Selenium (Se), a beneficial element in plants, can regulate plant growth by enhancing carbon metabolism and reducing heavy metal uptake. This study aimed to elucidate the protective mechanisms of Se application in licorice plants exposed to 20 μM Cd. Experiments with 1 and 5 μM of Se revealed that 1 μM of Se provided the best protective effects. This concentration reduced the Cd^2+^ content in the roots of *G. uralensis*, while significantly increasing plant biomass, root length, SPAD value, and contents of K^+^, Ca^2+^, and S^2−^. Additionally, the treatment reduced the malondialdehyde (MDA) content by 30.71% and 58.91% at 12 h and 30 d, respectively. The transcriptome analysis results suggest that Se mitigated Cd toxicity by enhancing carbon metabolism, regulating the AsA-GSH cycle, reducing Cd absorption, promoting Cd transport and compartmentalization, and modulating Cd resistance-associated transcription factors. These findings clarify the mechanisms by which Se alleviates Cd toxicity in *G. uralensis* and offer a promising strategy for the safe cultivation and quality control of medicinal herbs in Cd-contaminated soils.

## 1. Introduction

Cadmium (Cd) toxicity is one of the most severe forms of heavy metal pollution in soil, posing significant environmental and health risks owing to its high toxicity and carcinogenicity through the food chain [1,2,3]. Soil Cd pollution has become a critical global issue, with many countries, including China, facing serious contamination risks [4,5]. In China, approximately 7% of soil exceeds the Cd environmental quality standard, and this widespread distribution renders Chinese medicinal materials vulnerable to Cd contamination during cultivation [6,7]. *Glycyrrhiza uralensis* Fisch. (*G. uralensis*), known as “red-skinned licorice,” is a traditional medicinal plant in East Asia with various pharmacological properties, including antioxidant, antibacterial, neuroprotective, and anticancer effects [8,9]. It plays an irreplaceable role in traditional Chinese medicine prescriptions by harmonizing other herbs [8,10]. However, owing to the sharp decline in wild resources, *G. uralensis* has been listed as a second-level protected plant in China [11]. *Glycyrrhiza uralensis* is mainly distributed in the temperate desert and temperate grassland areas of arid northwestern China [12]. Artificial cultivation is an effective solution to address the shortage of wild *G. uralensis* resources. However, Cd pollution in the soil poses safety challenges for *G. uralensis* cultivation. In addition, the 2020 edition of the “Chinese Pharmacopoeia” limits the Cd content of licorice to 1 mg/kg [13]. Studies have shown that Cd levels in licorice plants can exceed this standard [14]. Cd stress disrupts normal plant development, impairs photosynthesis and root activity [15,16], and causes oxidative damage by generating toxic reactive oxygen species (ROS) and increasing malondialdehyde (MDA) levels, ultimately reducing the plant yield and quality [15,17]. Our previous investigations revealed that Cd stress inhibited *G. uralensis* growth and development, leading to Cd accumulation in the roots, reduced root growth, and decreased biomass [9]. Because the root is the primary medicinal component of licorice, it is imperative to identify effective strategies to reduce Cd accumulation in licorice roots, enhance Cd tolerance, and mitigate Cd toxicity.

Cd is absorbed by plant roots and compartmentalized in vacuoles as well as through xylem and phloem loading processes [18]. The “uptake” proteins responsible for transporting heavy metals from the external environment to the cytoplasm include natural resistance-associated macrophage proteins (NRAMPs) and zinc- and iron-regulated transporter-like proteins (ZIPs), which are typically located on the cytoplasmic membrane and facilitate Cd influx into the root cells. *OsNramp5* is a key member of the NRAMP family and is vital for regulating Cd and Mn uptake in rice root cells [19]. In addition to mediating iron and zinc uptake and transport, ZIP family proteins are involved in promoting Cd absorption by the plant roots [1,20]. To maintain tolerable Cd concentrations, plants have evolved various efflux mechanisms that transport heavy metals from the cytoplasm to the xylem, vacuoles, and other compartments. The heavy metal-associated (HMA) protein family is central to this efflux process, using ATP pump hydrolysis to generate energy for transporting heavy metal ions, such as Cd^2+^, Pb^2+^, Cu^2+^, and Zn^2+^, across membranes [21]. For example, *OsHMA2* mediates the efflux of Cd from xylem parenchyma cells into the xylem, and its knockout reduces Cd loading in the xylem and its translocation from the roots to aerial parts [22]. Moreover, increased expression of *OsHMA3* enhances vacuolar Cd sequestration in roots and improves Cd tolerance in rice [23]. Other protein families, such as the metal tolerance protein (MTP) family and yellow stripe-like transporter (YSL) family, also play crucial roles in plant responses to Cd stress [24,25,26].

The application of environmentally friendly materials for managing Cd pollution has increased significantly in recent years [27,28,29], with selenium (Se) emerging as a prominent area of research [30]. Se is a beneficial nutrient that supports plant growth and metabolism while enhancing plant tolerance to both biotic and abiotic stresses [31,32,33,34]. Research has indicated that Se can significantly reduce Cd accumulation in plant tissues [35], enhance root nutrient uptake [16,36], and mitigate Cd toxicity through various mechanisms. First, Se slows Cd-induced chlorophyll degradation, improves photosynthesis [16], enhances carbohydrate metabolism, and promotes plant growth [37,38]. Second, Se application increases the accumulation of non-enzymatic antioxidants by regulating the AsA-GSH cycle system [33,39,40], reduces lipid peroxidation, and strengthens the ability of plants to scavenge ROS [41], thereby minimizing Cd-induced oxidative damage [30]. Furthermore, the competitive relationship between Cd and Se in metal ion channels, particularly divalent ion channels [36,42], can contribute to the protective effect of Se by influencing Cd absorption, translocation, and accumulation in plant tissues [42,43,44].

Despite growing research on Se mitigation of Cd toxicity, its effects and mechanisms in licorice remain unclear. This study aimed to (1) explore the effects of Se treatment on the growth, nutrient uptake, and antioxidant capacity of *G. uralensis* seedlings under Cd stress, (2) determine whether Se application could inhibit Cd absorption and reduce Cd accumulation in *G. uralensis* roots, and (3) combine physiological and transcriptome data to identify the key differential pathways and transcription factors involved in the Se-mediated response to Cd stress using the WGCNA, Mfuzz, and integrated analysis. These goals aim to deepen our understanding of how Se influences the response of licorice to Cd stress and provides a scientific basis for developing safe cultivation and quality control strategies for medicinal herbs in Cd-contaminated soils.

## 2. Materials and Methods

### 2.1. Plant Materials and Experimental Design

The experimental material used in this study consisted of *Glycyrrhiza uralensis* Fisch. seeds, which were obtained from the Licorice Research Institute of Shihezi University. To break dormancy, the seeds were treated with 85% H_2_SO_4_ for 30 min, disinfected with 0.1% HgCl_2_ for 10 min, and rinsed 3–5 times with sterile water. The seeds were then soaked in sterile water for 4 h to absorb moisture and subsequently sown in bowls filled with vermiculite at a depth of approximately 1 cm, with three seeds per bowl. The bowls were placed in 1 L hydroponic boxes containing Hoagland nutrient solution for cultivation. Once the seedlings developed two true leaves (21 days), six treatment groups were established in the nutrient solution: CK (deionized water), Se1 (1 μM Na_2_SeO_3_ solution), Se5 (5 μM Na_2_SeO_3_ solution), Cd (20 μM CdCl_2_·2.5H_2_O solution), Cd + Se1 (20 μM CdCl_2_·2.5H_2_O and 1 μM Na_2_SeO_3_ solution), and Cd + Se5 (20 μM CdCl_2_·2.5H_2_O and 5 μM Na_2_SeO_3_ solution). Each treatment was replicated three times. The cultivation was conducted in a light incubator (GXZ-430D) under a 14 h light cycle at 28 °C and a 10 h dark cycle at 22 °C, with a light intensity of 280–420 μmol m^−2^ s^−1^. The nutrient solution was replaced every 3 d. Samples were collected at 12 h and 30 d after the start of treatment. The harvested seedlings were divided into two batches, one batch of samples was used for physiological parameter measurement and another batch of samples was used for RNA sequencing analysis. Samples of each treatment had three biological replicates.

### 2.2. Measurement of Plant Growth Parameters

At 30 d after treatment, ten seedlings were randomly selected from each treatment group, ensuring ten replicates per group for parameter measurements. The root length was measured for seedlings in the CK, Se1, Se5, Cd, Cd + Se1, and Cd + Se5 groups. The surface moisture was removed using absorbent paper, and the seedlings were placed in an oven at 105 °C for 15 min before being dried at 70 °C until reaching a constant weight. The dry weights of the aerial parts and roots were measured to determine biomass.

Chlorophyll content was estimated using a portable SPAD meter (Konica Minolta, Tokyo, Japan), which could provide a non-destructive method to assess leaf chlorophyll by analyzing its optical properties. Higher SPAD values indicated higher chlorophyll concentrations. For each plant, five measurements were taken from different leaves, and the average SPAD value was calculated to represent the chlorophyll content.

### 2.3. Determination of K^+^, Ca^2+^, Fe^2+^, Mg^2+^, S^2−^, Se^2+^, and Cd^2+^ Concentrations

At 30 d after treatment, *G. uralensis* roots were collected and digested following the method described by Li et al. [45], using a CEM MARS6 microwave digestion system with high-purity concentrated HNO_3_ and H_2_O_2_ in a 9:1 ratio. The concentrations of K^+^, Ca^2+^, Fe^2+^, Mg^2+^, S^2−^, Se^2+^, and Cd^2+^ in root samples were measured using an inductively coupled plasma optical emission spectrometer (ICAP-6300, Thermo Fisher Scientific, Waltham, MA, USA).

### 2.4. Assessment of MDA Levels

The roots of *G. uralensis* seedlings were collected at 12 h and 30 d. Fresh root samples (0.5 g) were weighed and homogenized in 5 mL of pre-cooled 5% trichloroacetic acid (TCA) solution using a mortar kept on ice. The homogenate was centrifuged at 2795× *g* for 10 min at 4 °C, and the clarified supernatant was collected for analysis. Malondialdehyde (MDA) levels were determined by reacting with thiobarbituric acid (TBA) and measuring absorbance at 532 nm, using the thiobarbituric acid reactive substances (TBARSs) assay. TBARS values were corrected for non-MDA interference by subtracting the 532 nm absorbance of samples incubated without TBA (-TBA controls), as described by Hodges’ method [46]. Absorbance was measured at 532, 600, and 450 nm using a UV spectrophotometer, and MDA concentration was calculated using the following formula:MDA (µmol·L^−1^)  =  6.45(D_532_ − D_600_) − 0.56D_450_.

### 2.5. Assessment of GSH Levels

The roots of *G. uralensis* seedlings were collected at 12 h and 30 d after treatment. For GSH determination, 0.1 g of root tissue was homogenized in 1 mL of extraction buffer on ice. The homogenate was centrifuged at 7155× *g* for 10 min at 4 °C, and the supernatant was collected for further analysis. GSH concentration was quantified using a glutathione assay kit (Solarbio^®^, Beijing Solarbio Science & Technology Co., Ltd., Beijing, China) according to the manufacturer’s instructions.

### 2.6. Total RNA Isolation and Illumina Sequencing

RNA sequencing of *G. uralensis* root samples from the CK, Cd, and Cd + Se1 treatment groups was performed at 12 h and 30 d. Each treatment was subjected to three biological replicates, and a total of 18 root samples were obtained for RNA sequencing, about 0.1 g per sample. Total RNA was extracted from root tissues using the TRIzol reagent kit (Thermo Fisher Scientific, USA), and the RNA concentration was measured using the Qubit RNA HS Assay Kit (Thermo Fisher Scientific, USA). RNA integrity was assessed using an RNA Nano 6000 analysis kit (Thermo Fisher Scientific, USA). A cDNA library was constructed using the NEBNext^®^ Ultra™ RNA kit (New England Biolabs, Ipswich, MA, USA), and the library fragments were purified using the AMPure XP system (Beckman Coulter, Beverly, CA, USA). Sequencing was conducted on the Illumina HiSeq™ platform and the resulting high-quality clean reads (evaluated for GC content, Q20, and Q30) were aligned to the licorice reference genome using HISAT2 (version 2.0.4) [47].

### 2.7. Differentially Expressed Gene Analysis

Differential gene expression across multiple groups was analyzed using DESeq2 [48], with genes classified as differentially expressed genes (DEGs) if |log2FoldChange| > 1 and *p* < 0.05. DEGs with FPKM > 1 were screened, and the co-expressed gene modules were identified through fuzzy clustering using the Mfuzz package (R environment) with the cluster quantity parameter set to five. Weighted gene co-expression network analysis (WGCNA) was conducted in R using the WGCNA package, applying standard settings to DEGs that met the expression threshold (FPKM > 1). GO and KEGG enrichment analyses were performed on the DEGs in the filtered modules, and combined with Mfuzz analysis to create a co-expression network diagram.

To validate the transcriptome sequencing results, we selected seven DEGs associated with Cd transmembrane transport for quantitative real-time PCR (qRT-PCR) verification. To ensure reliability, each sample was analyzed three times biologically. The *Glycyrrhiza inflata* Bat. 18S rRNA was used as the reference gene, and the relative expression levels of target genes were calculated by the 2^−ΔΔCt^ method. All primers used for qRT-PCR are listed in Appendix A. Real-time quantitative PCR analyses were carried out by Shanghai Biozeron Biotechnology Co., Ltd. (Shanghai, China).

### 2.8. Statistical Analysis

Data were presented as the mean ± standard error of at least three independent biological replicates. Dry weight, root length, SPAD value, MDA content, GSH content, and ion content were analyzed using Microsoft Excel and ANOVA in R. Significant differences were assessed by the Tukey test (Tukey’s HSD function in the stats package).

## 3. Results

### 3.1. Effects of Se Application on the Growth of G. uralensis Seedlings Under Cd Stress

The experimental data showed that Cd exposure significantly reduced biomass accumulation in *G. uralensis* roots, but this effect was alleviated by the application of Se. Compared with the Cd treatment group, the Cd + Se1 and Cd + Se5 groups showed 77.4% and 50.94% increases in the dry weight of *G. uralensis* roots (Figure 1B). Additionally, Se application significantly increased root length and SPAD values under Cd stress (Table 1 and Table 2). Notably, Cd treatment inhibited biomass accumulation in the aerial parts of *G. uralensis* compared with that in the CK group (Figure 1A), and the Cd + Se1 treatment increased the aerial biomass (Figure 1A).

### 3.2. Effects of Se Application on Cd and Mineral Element Absorption of G. uralensis Seedlings Under Cd Stress

The results showed that Se1 increased Se^2−^ content in the roots of *G. uralensis* under Cd exposure but had no effect on Se^2−^ levels in the aboveground parts, indicating efficient utilization of Na_2_SeO_3_ in the roots (Figure 2A). Cd accumulation was significantly higher in the roots than in the shoots. Se1 reduced Cd^2+^ accumulation in the roots by approximately 27% but did not significantly affect Cd^2+^ levels in the shoots (Figure 2B). In contrast, Se5 significantly decreased the Cd^2+^ concentration in the shoots but had no notable effect on the Cd^2+^ content in the roots (Figure 2B).

Se positively influenced the accumulation of mineral elements in *G. uralensis* under Cd stress, with the Se1 treatment proving more effective (Figure 2). Cd exposure reduced the K^+^, Ca^2+^, S^2−^, and Fe^2+^ levels, while increasing the Mg^2+^ content in the aerial parts. In the roots, Cd decreased the K^+^, Ca^2+^, and S^2−^ levels but increased the Fe^2+^ and Mg^2+^ content (Figure 2C–G). Under Cd stress, 1 μM of Se increased the K^+^, Ca^2+^, Mg^2+^, and S^2−^ levels in the aerial parts by 25.39%, 19.79%, 14.46%, and 24.97%, respectively, while reducing the Fe^2+^ content by 31.05%. In contrast, 5 μM of Se increased the K^+^ content by 10.79% and reduced the Fe^2+^ levels by 36.05% but had no significant effect on the Ca^2+^, Mg^2+^, and S^2−^ levels. In roots, both 1 and 5 μM Se treatments had no significant effect on the K^+^, Ca^2+^, and Mg^2+^ levels but increased the S^2−^ content by 34.12% and 25.83% and reduced the Fe^2+^ levels by 26.15% and 34.46%, respectively (Figure 2C–G).

### 3.3. Effects of Se Application on MDA and GSH Contents in G. uralensis Seedlings Under Cd Stress

Analysis of biomass, Cd^2^^+^ content, and mineral element levels in *G. uralensis* revealed that Se1 was more effective than Se5 in reducing Cd^2+^ accumulation in roots. To better understand the slow-release mechanism of Se, further experiments were conducted using 1 μM of Se.

The results indicate that Se1 treatment did not affect MDA levels in *G. uralensis* roots, whereas Cd stress significantly increased MDA content at both 12 and 30 d. However, the Cd + Se1 treatment effectively reduced MDA levels by 30.71% and 58.91% at 12 and 30 d, respectively (Figure 3A,B). GSH levels in the roots of *G. uralensis* seedlings significantly increased under Cd stress at both time points, with no significant effect of Se application. Additionally, Se1 treatment increased GSH content at 12 h but had no effect within 30 d (Figure 3C,D).

### 3.4. Overview of DEGs

Transcriptomic analysis was conducted on the Cd20 and Cd20 + Se1 treatment groups at 12 h and 30 d, with the quality control data presented in Appendix A. At 12 h, 30,286, 30,428, and 30,306 DEGs were identified in the Cd20_Se0 vs. Cd0_Se0, Cd20_Se1 vs. Cd0_Se0, and Cd20_Se1 vs. Cd20_Se0 groups, respectively, with 1275, 1780, and 861 upregulated and 1656, 1249, and 405 downregulated by 30 d; 30,375, 30,340, and 30,208 DEGs were identified in the same groups, with 5208, 3890, and 1667 upregulated and 5444, 3290, and 1587 downregulated, respectively (Figure 4).

### 3.5. Mfuzz Analysis

Mfuzz clustering and GO analysis of differential genes from the six comparison groups identified five gene clusters (C1–C5) with distinct expression trends and annotation information. Groups C3 and C4 presented high expression scores (z-scores) for Cd20_Se0 and Cd20_Se1 at 12 h and 30 d, whereas groups C1, C2, and C5 exhibited lower overall expression scores. In the C3 and C4 clusters, GO terms, such as cellular metabolic processes, ribosomal structural components, ATP metabolic processes, and carbohydrate metabolic processes, were significantly enriched. In the C1, C2, and C5 clusters, GO terms, such as ion binding, protein modification processes, regulation of DNA templating, and ribosomes were significantly enriched (Figure 5A). KEGG enrichment analysis revealed the top three upregulated and downregulated annotation terms, demonstrating significant enrichment for pathways such as starch and sucrose metabolism, oxidative phosphorylation, alanine–aspartate–glutamate metabolism, α-linolenic acid metabolism, and flavonoid metabolism across the treatment comparison groups (Figure 5B).

### 3.6. WGCNA Analysis

WGCNA analysis of DEGs identified 14 gene-phenotype co-expression modules, among which the turquoise module was significantly positively correlated with biomass (*p* < 0.05). The gene clusters in the yellow and blue modules exhibited similar expression patterns, as reflected by consistent coloration in the correlation heatmap (Figure 6A,C), and both were significantly negatively correlated with biomass (*p* < 0.05). These findings suggest that the three modules captured the core physiological processes and their associated gene clusters in *G. uralensis* seedling tissues. Therefore, the turquoise module and combined “yellow + blue module” were selected for further analysis. APEAR analysis was performed on the two module gene sets. The network diagram illustrates that in the “turquoise” module, GO terms, including protease core complex, phosphate nucleoside biosynthesis process, and peptide chain complex endonuclease, were enriched (Figure 7A). In the “yellow + blue” module, significant enrichment was observed in the GO terms such as MAPK cascade amplification reaction and discarded thylakoids (Figure 7B).

### 3.7. Analysis of Mfuzz Combined with WGCNA

GO enrichment analysis of the “turquoise” and “yellow + blue” modules (*p* < 0.05) identified significant terms for each module. In the “turquoise” module, the key terms included the MF terms threonine endopeptidase activity and primary active transmembrane transporter activity, the CC terms proteasome core complex and endopeptidase complex, and the BP terms nuclear coenzyme-containing small molecule metabolism and discarded cell macromolecule catabolism. For the “yellow + blue” module, the significant terms included the MF terms ADP binding and nucleic acid binding, CC terms thylakoid and photosystem, and BP terms photosynthesis and cellular processes (Figure 8). The joint analysis of Mfuzz and WGCNA pairing the annotations with network links revealed that C3 and C4 were co-matched with the “turquoise” module, while C1, C2, and C5 were co-matched with the “yellow + blue” module (Figure 9). This alignment between the Mfuzz trend analysis and the gene–trait co-expression analysis suggests distinct characteristics of shared enrichment pathways. For instance, C3 and C4 modules associated with the "turquoise" module demonstrated upregulated gene expression in response to Cd and Cd + Se1 treatments compared to CK, indicating their activation under stress conditions (Figure 5). Additionally, the significant enrichment of these pathways in the “turquoise” module suggests a positive association with the growth of *G. uralensis* seedlings (Figure 6).

The KEGG enrichment analysis of the “turquoise” and “yellow + blue” module gene sets was combined with the KEGG enrichment results from different comparison groups, revealing the significant enrichment in the pathways related to amino acid metabolism, ascorbic acid and aldehyde acid metabolism, fatty acid synthesis and degradation, and secondary metabolism (including flavonoid biosynthesis, α-linolenic acid metabolism, and phenylpropanoid biosynthesis) across modules and treatment groups (Figure 9A). Further matching of the “turquoise” and “yellow + blue” module gene sets with the KEGG enrichment annotations of the C1–C5 trend gene set indicated that the mRNA surveillance pathway was significantly enriched in C5/blue. Meanwhile, pathways such as peroxisome, one-carbon pool by folate, glutathione metabolism, riboflavin metabolism, arachidonic acid metabolism, and terpenoid backbone biosynthesis were significantly enriched only in C3 and C4/turquoise (Figure 9B).

### 3.8. Identification of Key Pathways and Genes

#### 3.8.1. DEGs Involved in Carbon Metabolism

DEGs were enriched in C metabolic pathways, including alanine, aspartic acid, glutamic acid metabolism, starch and sucrose metabolism, fatty acid degradation, and oxidative phosphorylation (Figure 10). Under Cd treatment for 30 d, the expression of genes such as *ASSY*, *ASPGA*, *ARLY*, and *GAD5* in alanine, aspartate, and glutamate metabolism was downregulated compared to that in normal plants, whereas Se application significantly upregulated these genes. Similar results were observed for starch and sucrose metabolism (*PHS1*, *DPE1*, *PGMP*, *ISA2*, *ISA3*, *SS3*, *BGLU431*, *BGLU44*, and *BGLU46*), fatty acid degradation (*ADHL7*, *ADHL1*, *Q9FH04*, and *ALDH2B7*), and oxidative phosphorylation pathways (*ATP5E*, *CIB22*, *NDUA6*, *SDH3-1*, *PPA6*, *FRO1*, and *PPA1*). These findings suggest that the 30-day Cd stress reduced the activity of these metabolic pathways, while the Se application reactivated them. Notably, the genes in these pathways were significantly downregulated under the 12 h Cd + Se1 treatment.

#### 3.8.2. DEGs Involved in Cd Transport

These findings revealed that Cd treatment and Se application influenced the expression of Cd-transport-related protein genes (Figure 11). At 12 h, Cd stress upregulated the expression of *YSL1*, *YSL3*, *ZIP2*, *HMA3*, *MTP4*, and *MTP10*. Se application reduced the expression of *YSL1*, *YSL3*, *ZIP2*, and *HMA3* while enhancing *MTP4* and *MTP10*. At 30 d, Cd stress also upregulated *YSL7*, *IRT3*, *ZIP6*, *ZIP10*, *NRAM3*, *NRAM5*, *HMA2*, *HMA4*, and *HMA5*, whereas Cd + Se1 treatment downregulated these genes. By 30 d, Cd treatment upregulated *MTP1* expression, which was further enhanced by Se, while downregulating *MTP11* and *MTPB*, whose expression was restored by Se application. Moreover, Cd stress significantly downregulated *CAX1* expression at both 12 h and 30 d, whereas the Cd + Se1 treatment alleviated this effect (Figure 11B). These results suggest that Se application regulated the expression of Cd transmembrane transporter genes, thereby influencing Cd absorption and transport.

#### 3.8.3. DEGs Involved in AsA-GSH

KEGG enrichment analysis revealed significant enrichment in the glutathione metabolism and ascorbic acid–aldehyde acid metabolism pathways involving multiple DEGs (Figure 12). After 12 h of Cd stress, glutathione metabolism was promoted by the upregulated expression of *GSTL1*, *GSTL3*, *GSTF6*, *GSTU7*, *GSTU8*, *GSTU10*, *GSH1*, *GSH2*, *PGD1*, *PGD2*, *G6PD6*, *ICDHP*, and *GSHRC*, whereas ascorbic acid metabolism was inhibited by the downregulated expression of *ALDH2B7*, *UGD3*, *UGD4*, *GME*, *APX3*, *VTC4*, and *LGALDH*. Se application enhanced both metabolic pathways induced by Cd, with a bias towards glutathione metabolism (Figure 12). After 30 d of Cd stress, glutathione metabolism genes (*GSTL3*, *GSTU10*, *GSH1*, *GSH2*, *G6PD6*, *RNR2A*, *ICDHP*, and *GSHRC*) were upregulated, whereas ascorbic acid metabolism genes (*MIOX1*, *ALDH7B4*, *ALDH2B7*, *UGD3*, *UGD4*, *GME*, *VTC2*, *VTC4*, and *APX3*) were downregulated. However, the Cd + Se1 treatment reversed the expression levels of these DEGs (Figure 12). Long-term Cd stress disrupted the balance between ascorbic acid and glutathione metabolism, favoring glutathione metabolism, while Cd + Se1 treatment shifted the ASA-GSH cycle balance towards ascorbic acid metabolism.

#### 3.8.4. Expression of Transcription Factors

In the “turquoise” module, which was significantly positively correlated with biomass, all transcription factors were analyzed. Under long-term Cd stress (30 d), most transcription factors associated with Cd accumulation and tolerance in plants were upregulated. Owing to its antagonism with Cd, Se application downregulated transcription factor expression (Figure 13). Furthermore, Cd stress inhibited the expression of certain transcription factors, such as Glyur000975s00029570, Glyur000821s00020796, and Glyur000132s00015923, whereas Se application enhanced their expression (Figure 13).

#### 3.8.5. qRT-PCR Validation of DEG Results

To validate the accuracy of the transcriptome sequencing results, we randomly selected seven DEGs related to cadmium uptake, compartmentalization, and transport for qRT-PCR verification. The results demonstrated that the expression patterns of these seven DEGs were consistent with the transcriptome sequencing data (Appendix A).

## 4. Discussion

### 4.1. Se Promotes the Growth of G. uralensis Seedlings Under Cd Stress

Previous studies have shown that Cd can negatively affect plant growth and inhibit early seedling development, with toxicity evident in both morphological and physiological aspects [49]. In this study, *G. uralensis* seedlings exhibited significantly reduced biomass, root length, and SPAD values under Cd stress. However, the application of Se improved these parameters, with the biomass in roots and aboveground parts recovering to higher levels, and root length and SPAD values increasing significantly (Figure 1; Table 1 and Table 2), which was consistent with previous findings [16]. The increase in SPAD values indicated a higher chlorophyll content [50], suggesting that Se may enhance the photosynthetic potential of *G. uralensis* seedlings [51]. Under Cd stress, the K^+^ and Ca^2+^ contents in the aboveground tissues of *G. uralensis* decreased (Figure 2C,D). Treatment with 1 μM of Se increased K^+^ and Ca^2+^ levels by 25.39% and 19.79%, respectively, whereas treatment with 5 μM of Se had no significant effect on Ca^2+^ and increased K^+^ content by 10.79%. Additionally, Cd exposure reduced S^2−^ levels in *G. uralensis* roots, whereas the 1 and 5 μM Se treatments increased S^2−^ content by 34.12% and 25.83%, respectively (Figure 2C,D,F). These results indicate that Se could alleviate the inhibitory effects of Cd stress on K, Ca, and S absorption, with the mitigation effects varying with Se concentration. Although low Se levels promote plant growth, high concentrations can have inhibitory effects [52,53]. In this study, 1 μM of Se effectively mitigated the negative effects of Cd on biomass, root length, and SPAD value, with significantly better results than those of the 5 μM Se treatment (Figure 1; Table 1 and Table 2).

### 4.2. Se Regulates Genes Involved in the Carbon Metabolism in G. uralensis Seedlings Under Cd Stress

Carbon metabolism is essential to provide the energy required for plant growth and development. Cd stress can significantly inhibit plant growth by disrupting carbohydrate absorption and suppressing carbon metabolism, leading to reduced biomass [54,55]. Se can enhance plant growth and biomass accumulation by positively affecting carbohydrate metabolism [37,38]. In this study, the genes involved in starch and sucrose decomposition, such as *DPE1*, *DPE2*, *PHS2*, *PGMP*, and *ISA3*, were significantly upregulated after 30 d of Cd + Se1 treatment (Figure 11). Additionally, the tricarboxylic acid (TCA) cycle and oxidative phosphorylation, which are key metabolic pathways in cellular respiration, were promoted by Cd + Se1 treatment at 30 d. Genes related to alanine, aspartate, and glutamate metabolism and fatty acid degradation pathways were significantly upregulated under the same conditions, enhancing the TCA cycle (Figure 10). These pathways can play vital roles in plant responses to abiotic stresses [56,57] because glutamine and aspartate generate α-keto acids through deamination, which serve as intermediates in the TCA cycle. Fatty acids are converted to acetyl-CoA through β-oxidation, which provides substrates for the TCA cycle [58]. Furthermore, genes associated with oxidative phosphorylation, such as *ATP5E*, *CIB22*, and *NDUA6*, were also upregulated under Cd + Se1 treatment at 30 d, promoting oxidative phosphorylation. Enhanced TCA cycle and oxidative phosphorylation generate large amounts of ATP and NADH, supporting cellular respiration and providing sufficient energy to sustain plant growth. This energy boost likely contributes to the alleviation of biomass reduction caused by Cd stress [37,59]. However, at 12 h, Se did not restore the transcriptional levels of many genes affected by Cd in carbon metabolism, indicating that Se application had limited effects on *G. uralensis* carbon metabolism during the early stages of Cd stress (12 h).

### 4.3. Se Improves Cd Tolerance in G. uralensis Seedlings by Regulating Genes Involved in Transport Proteins

Metal transporters are critical for metal uptake and transport in plants. As a non-essential element, Cd lacks a specific transport pathway in plants and competes with other divalent ion channels for transport [29,60].

This study revealed that Cd accumulation in *G. uralensis* roots increased significantly under Cd exposure, whereas Se1 application reduced Cd^2+^ accumulation in roots by approximately 27% (Figure 2B). The transcriptome data indicated that Se application regulated Cd accumulation in *G. uralensis* seedlings by enhancing the efficiency of Cd^2+^ movement through other ion channels, particularly divalent ion channels such as Fe, Ca, or Mg. To better understand the roles of related genes in managing Cd, their functions were categorized into four main categories: (1) Cd uptake from the external environment, (2) Cd storage in vacuoles, (3) bidirectional transport of Cd between xylem vessels and parenchymal cytoplasm, and (4) long-distance transport of Cd from roots to shoots.

Reducing Cd absorption from the external environment is critical for mitigating Cd accumulation in plants and minimizing its toxic effects [60,61,62]. NRAMP and ZIP protein families play vital roles in this process. In rice, *OsNramp5* serves as a key transporter in the NRAMP family, facilitating Cd uptake from the external environment into root cells [19], and knockout of the *OsNramp5* gene significantly reduces Cd accumulation and plant growth [63]. Similarly, the *SaNramp6* gene expressed in the plasma membrane of epidermal cells in transgenic Arabidopsis can increase Cd accumulation in plants [64]. Studies have indicated that *OsZIP1* and *AtZIP10* are involved in Cd^2+^ absorption in the roots of rice and Arabidopsis, respectively [65,66]. Silencing of the *ZIP6* gene can reduce Cd uptake in Arabidopsis and enhance root Cd resistance [67]. This study revealed that Cd stress upregulated the expression of *NRAMP5*, *NRAMP6*, *ZIP1*, *IRT3*, *ZIP6*, and *ZIP10* genes by 30 d, whereas the Cd + Se1 treatment reduced their expression (Figure 11B). These findings suggest that Se application decreased Cd uptake from the external environment in *G. uralensis* roots, thereby alleviating Cd toxicity (Figure 11A).

The Cd storage in vacuoles is a key compartmentalization strategy at the plant cell level to reduce the Cd^2+^ toxicity in the cytoplasm [68,69]. As members of the cation diffusion facilitator (CDF) family, Metal Tolerance Proteins (MTPs) play a pivotal role in transporting Cd^2^^+^ from the cytoplasm into vacuoles, maintaining cellular homeostasis during metal stress [24]. Heterologous expression of *MTP1* in tobacco increased vacuolar thiol content and enhanced Cd tolerance, reducing the adverse effects of Cd on plants [24]. Similarly, *OsMTP11* facilitates Cd sequestration in the vacuoles of leaf vascular cells, limiting its movement into rice grains [70]. In this study, *MTP4* and *MTP10* expression was upregulated after 12 h of Cd stress, and Cd + Se1 treatment further enhanced their expression. After 30 d, Cd stress upregulated *MTP1* expression, which was further increased by Se application, whereas Cd stress downregulated *MTP11* and *MTPB*, and Se application restored their expression (Figure 11B). Additionally, *CAX1* is a calcium ion transport protein localized in the vacuole membrane, is involved in heavy metal ion translocation (e.g., Mn^2+^ and Cd^2+^) [71], and can be positively correlated with Cd tolerance [71,72]. However, Cd treatment significantly downregulated *CAX1* expression at both 12 h and 30 d, whereas Cd + Se1 treatment alleviated this effect (Figure 11B). These findings suggest that *G. uralensis* can regulate the expression of MTP and CAX family proteins to enhance vacuolar Cd sequestration, thereby reducing Cd toxicity. Se application under both short-term (12 h) and long-term (30 d) conditions can continuously improve the vacuolar Cd sequestration capacity (Figure 11A).

The parenchyma cells in the stele play a crucial role in distributing Cd throughout plants by regulating the balance of Cd^2+^ levels between the cytoplasm and xylem vessels through Cd^2+^ efflux [29,68,73]. This process is largely dependent on the HMA protein family, which ensures a balanced and efficient flow of Cd. Previous studies have shown that *HMA2* and *HMA4* can transport Cd^2+^ from roots to shoots via the xylem, enhancing plant tolerance to Cd stress [74,75], whereas *OsHMA5* in rice is involved in the xylem loading of Cu [76]. In our study, the expression of *HMA2*, *HMA4*, and *HMA5* was upregulated after 12 h of Cd stress, and the Cd + Se1 treatment further enhanced their transcription levels (Figure 11B). This suggests that Se application regulated the expression of these genes, accelerating Cd transport between the cytoplasm of root stele parenchyma cells and xylem vessels, and balancing Cd xylem unloading and longitudinal transport [77] (Figure 11A). Notably, *HMA3* primarily mediated Cd transport into vacuoles rather than xylem efflux [78,79], and it was downregulated under Cd stress but upregulated under Cd + Se1 treatment by 30 d.

Cd promotes its distribution throughout various plant sections via longitudinal transport through the xylem from the roots to aerial parts, which can be enhanced by yellow stripe-like (YSL) family proteins [80,81,82]. YSL transporters facilitate the movement of metal ions complexed with plant siderophores or nicotinamide (NA) and regulate the absorption, internal transport, and allocation of mineral nutrients within plants [25]. Studies have indicated that the YSL family can participate in Cd absorption and long-distance transport from roots to aerial parts [83]. The overexpression of Miscanthus *MsYSL1* in Arabidopsis improves Cd tolerance by mediating the redistribution of metal ions [25], whereas *YSL3* located in root and stem vascular tissues can catalyze the transport of various metal–NA complexes, including Cd-NA, to enhance heavy metal tolerance [84,85]. After 30 d of Cd stress, our data revealed that *YSL1* expression was inhibited, whereas *YSL3* expression was upregulated. Se application further upregulated the transcription of both *YSL1* and *YSL3* (Figure 11B), indicating that Se promoted the rational distribution of Cd from roots to shoots by enhancing *YSL1* and *YSL3* expression, thereby reducing Cd toxicity in *G. uralensis* roots (Figure 11A).

These findings demonstrate that Se plays a crucial regulatory role in Cd^2+^ transport in *G. uralensis* seedlings. Se selectively enhanced the activity of transmembrane transporter-related genes at different stages (12 h and 30 d), regulating Cd uptake, sequestration, short-distance translocation, and long-distance transport in plant roots. This modulation helps to protect the physiological state and growth potential of plants under Cd stress, alleviating its harmful effects. These results highlight the multilevel regulatory mechanism of Se in plant Cd stress responses and offer new insights into improving plant Cd tolerance.

### 4.4. Se Regulates Genes Involved in the AsA-GSH Cycle to Reduce Oxidative Damage in G. uralensis

Cd stress can significantly increase ROS production in plants, causing plasma membrane peroxidation and organelle damage [60,86]. The addition of Se to a Cd solution can significantly inhibit the accumulation of H_2_O_2_ and MDA [62]. In this study, Cd stress increased MDA content in *G. uralensis* seedlings, indicating aggravated cell membrane damage, which served as an important factor inhibiting seedling growth under Cd stress. Se application reduced the MDA content to lower levels (Figure 3A,B), suggesting that Se alleviated the negative effects of Cd stress in plants. The AsA-GSH cycle is a critical antioxidant defense pathway for ROS detoxification under abiotic stress [87], with GSH and ascorbic acid (AsA) playing vital roles. Se can promote the synthesis of non-enzymatic antioxidants (GSH and AsA), thereby reducing oxidative stress in plants [88,89]. Transcriptome analysis showed that under the 12 h Cd stress with Se treatment, DEGs related to glutathione metabolism were upregulated, while those related to ascorbic acid metabolism were downregulated. After 30 d, the expression trends of these genes were reversed. These findings indicate that the *G. uralensis* seedlings adopted different Se utilization strategies during the short-term (12 h) stress response and long-term (30-day) adaptive response to Cd stress.

In the AsA-GSH cycle, GSH functions as an antioxidant by scavenging ROS [87]. The short-term (12 h) transcriptome data revealed that Cd stress upregulated genes associated with GSH synthesis, which was further amplified by Cd + Se1 treatment, including genes such as *GSH1*, *GSH2*, *PGD1*, *PGD2*, *G6PD6*, *ICDHP*, and *GSHRC* (Figure 12). Glutathione S-transferases (GSTs) serving as a multifunctional enzyme superfamily are critical for plant detoxification processes [90]. GSTs can catalyze the S-conjugation of GSH with toxic substances, and the resulting conjugates are sequestered into vacuoles or transferred to apoplasts to protect cells from chemical-induced toxicity and enhance tolerance [91]. Studies on Arabidopsis have shown that GSTs are tightly regulated through multiple independent signaling pathways and contribute to oxidative stress protection [92]. Additionally, under As stress, increased GSH content in plants has been linked to the elevated expression of genes encoding GSTs and GPX [93]. Similarly, this study observed that Cd + Se1 treatment at 12 h upregulated the expression of GST family genes, including *GSTL1*, *GSTL3*, *GSTF6*, *GSTU7*, *GSTU8*, and *GSTU10* (Figure 12). These findings suggest that short-term Se application enhances the expression of genes involved in GSH and GST synthesis, promotes the glutathione metabolic pathway, alleviates the oxidative stress caused by Cd stress, and actively supports the growth and development of *G. uralensis* seedlings.

A balanced metabolism is essential for improving plant productivity. Long-term (30-day) Cd stress disrupted the balance of the glutathione–ascorbic acid cycle by inhibiting ascorbic acid metabolism and promoting glutathione metabolism. Although increased glutathione metabolism partially compensated for the reduced ascorbic acid activity, it could be insufficient to maintain normal antioxidant defense mechanisms. Exogenous Se application repaired this disrupted cycle, promoting ascorbic acid metabolism through the regulatory participation of genes such as *MIOX1*, *ALDH7B4*, *ALDH2B7*, *UGD3*, *UGD4*, *GME*, *VTC2*, *VTC4*, and *APX3* (Figure 12). *MIOX1*, *UGD3*, *UGD4*, *GME*, *VTC2*, and *VTC4* are directly involved in ascorbic acid biosynthesis, and *MIOX1* and *UGD4* also contribute to UDP-glucuronic acid biosynthesis, which can supply nucleotide sugars for cell wall polymers [94]. Ascorbate peroxidase (APX) is a key ROS scavenger that reduces intracellular H_2_O_2_ accumulation, protects plants from oxidative stress, and improves tolerance to abiotic stress [95,96]. Additionally, ALDH enzymes remove harmful endogenous or exogenous aldehydes, such as reactive aldehydes, from lipid peroxidation or environmental pollutants, thereby promoting cellular antioxidant defense and maintaining normal cellular function and structural integrity [97]. The upregulated expression of these genes following Se application can help remove excess ROS under stress, restore metabolic balance, and enhance plant resilience.

### 4.5. Se Regulates the Expression of Transcription Factors Related to Cd Tolerance

Transcription factors can be crucial for regulating plant responses to heavy metals [98], including both their positive and negative effects on Cd accumulation and tolerance. For instance, the transcription factor *TaWRKY74* positively affects ASA-GSH synthesis genes and inhibits Cd transporter gene expression, thereby reducing Cd accumulation and toxicity in wheat [99]. Similarly, *MYB75*, *SpbZIP60*, and *TaNAC22* enhance Cd tolerance by regulating ROS homeostasis or upregulating Cd tolerance-related genes [100,101,102]. Transcription factors, such as *bHLHOBP3* and *PvERF15*, also improve Cd tolerance. *GmORG3* (a *bHLHOBP3* response gene) stabilizes Fe homeostasis, whereas *PvERF15* binds to the *PvMTF-1* promoter and its knockout reduces Cd tolerance [103,104]. However, some transcription factors can negatively affect Cd response, where Cd accumulation in plants increases and tolerance decreases. For example, *MYB49*, *TaMYC8*, and *AtWRKY12* are negative regulators. *MYB49* can directly bind to the promoters of *bHLH38* and *bHLH101*, positively regulating Cd uptake transporters and leading to higher Cd accumulation [105]. Similarly, *TaMYC8* can regulate the expression of *TaERF6* and activate *TaERF6* transcription, thereby increasing Cd accumulation in wheat [106]. *AtWRKY12* overexpression has been shown to increase Cd sensitivity in plants [107], whereas the C_2_H_2_ family protein *ZAT17* can negatively regulate plant tolerance to Cd [108]. In this study, the transcription factor families were positively associated with biomass; for example, the MYB family (Glyur000730s00036461, Glyur000975s00029570, and Glyur000986s00028882) was upregulated under 30-day Cd stress with Se application, thereby promoting plant growth. Conversely, the negative transcription factors (Glyur000001s00000030, Glyur000557s00018783, and Glyur000098s00007974) were downregulated by Se under the same conditions, alleviating their inhibitory effects on growth. Other transcription factor families with regulatory effects similar to those of the MYB family included WRKY, ERF, bHLH, C_2_H_2_, NAC, bZIP, and HSF (Figure 13). These findings suggest that Se modulates the expression of transcription factors, influencing downstream genes related to Cd uptake, translocation, and detoxification, thereby reducing Cd accumulation in *G. uralensis* seedlings and enhancing Cd resistance.

## 5. Conclusions

This study investigated the potential mechanism by which Se alleviated Cd toxicity in *G. uralensis* seedlings using physiological experiments and transcriptome sequencing. These results demonstrated that Se application promoted carbon metabolism, increased photosynthetic potential, and mitigated the adverse effects of Cd stress on seedling growth. Se enhanced the antioxidant capacity of *G. uralensis* by regulating genes involved in the AsA-GSH cycle pathway, effectively reducing cell membrane damage. Furthermore, Se treatment significantly decreased Cd accumulation in roots and reduced Cd toxicity. Transcriptomic analysis identified DEGs involved in Se-mediated Cd stress responses, including reduced Cd absorption, enhanced Cd sequestration in vacuoles, and optimized expression of transporters for Cd distribution in seedlings. Furthermore, Se modulated the expression of Cd-tolerant transcription factors, thereby improving Cd tolerance in *G. uralensis*. These findings elucidated the mechanisms by which Se reduced Cd accumulation and toxicity while highlighting the temporal differences in Se’s strategies for responding to Cd stress at 12 h and 30 d. This study suggests that Se application could be an effective strategy to promote the healthy growth of licorice plants and lower Cd content in licorice grown in Cd-contaminated areas.

## Figures and Tables

**Figure 1 plants-14-01736-f001:**
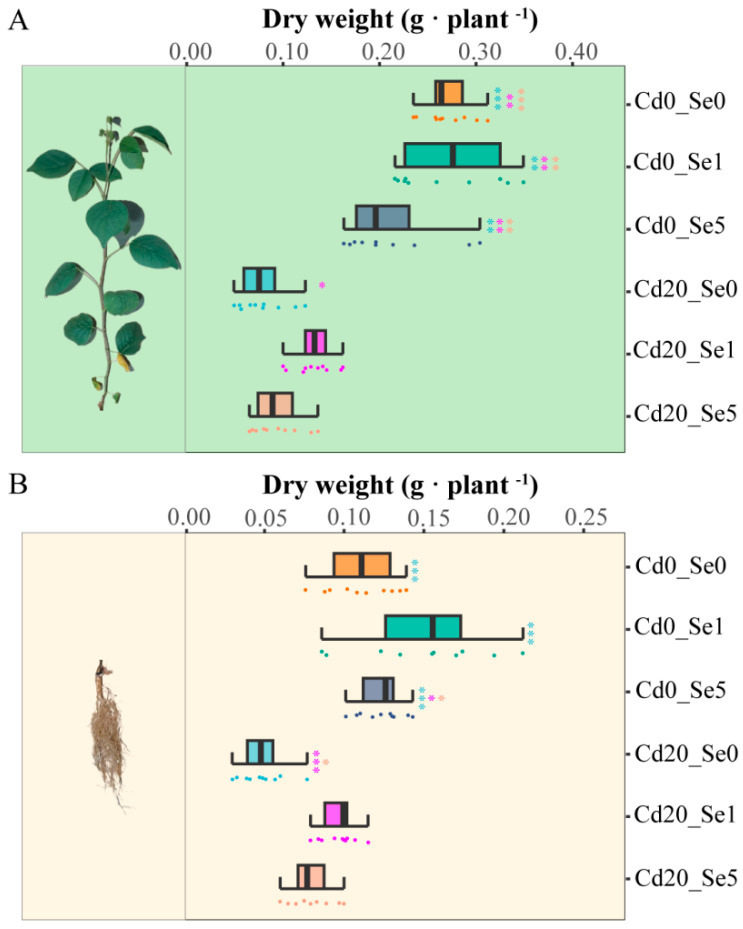
The biomass of the roots and shoots of *G. uralensis* on the 30th day of growth under different concentrations of Cd, Se, and Cd + Se (*n* = 10). (**A**) The dry weight of the shoot. (**B**) The dry weight of the root. Note: The difference analysis was performed using the R function *t*-test. The different colored markers beside each box plot represent that the group has differences from the treatment groups of box plots that have the same color as these markers. * *p*  <  0.05, ** *p*  <  0.01, and *** *p*  <  0.001.

**Figure 2 plants-14-01736-f002:**
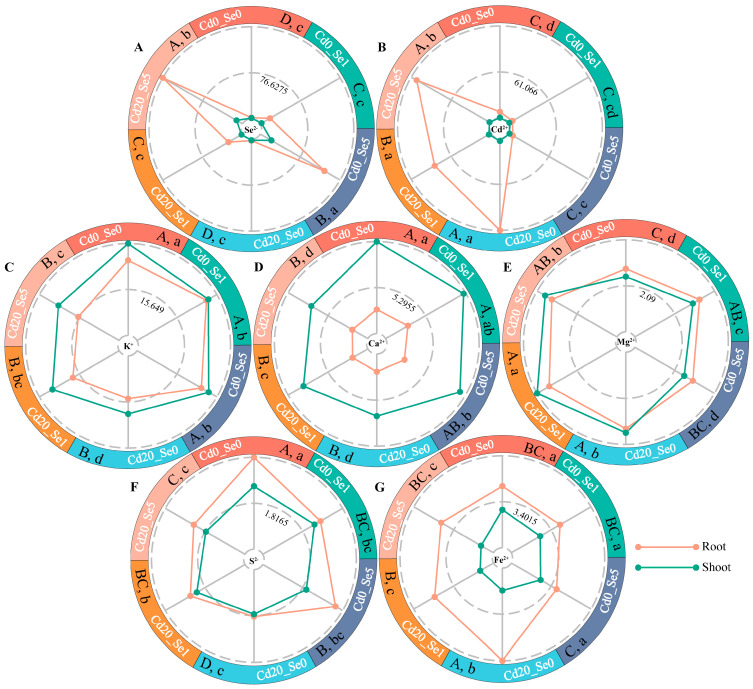
The contents of seven ions in the roots and shoots of *G. uralensis* at the 30th day of growth under different concentrations of Se, Cd, and Cd + Se (µg·g^−1^ DW). Subfigures (**A**–**G**) show: (**A**) Se^2−^, (**B**) Cd^2+^, (**C**) K^+^, (**D**) Ca^2+^, (**E**) Mg^2+^, (**F**) S^2−^, and (**G**) Fe^2+^. Note: Dotted lines and dots represent the mean ion contents in the roots and shoots (*n* = 3), and the difference analysis was conducted using Tukey’s HSD function. According to the Tukey test, different uppercase (lowercase) letters indicate significant differences in root (shoot) ion content between treatments (*p* < 0.05).

**Figure 3 plants-14-01736-f003:**
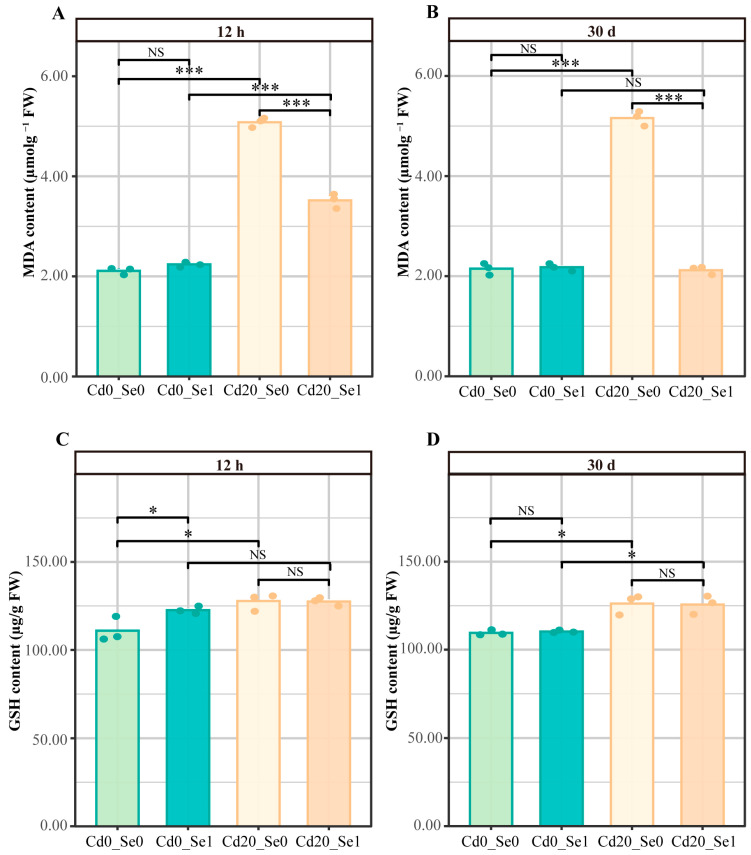
The contents of MDA and GSH in the roots of licorice at 12 h and 30 d under different concentrations of Se, Cd, and Se + Cd (*n* = 3). (**A**,**B**) MDA content. (**C**,**D**) GSH content. Note: The difference analysis was performed using the R function *t*-test. * *p * <  0.05 and *** *p*  <  0.001; “NS” means no significance.

**Figure 4 plants-14-01736-f004:**
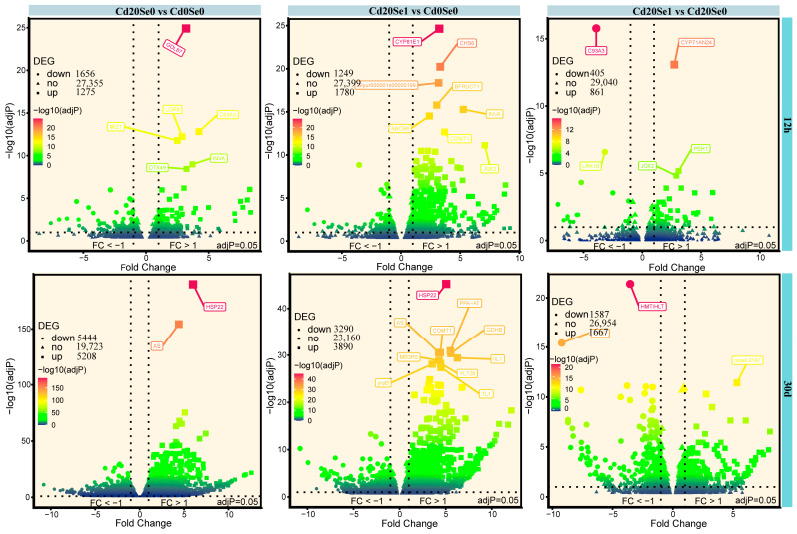
The difference volcano map of gene expression under the comparison of different treatment groups; the horizontal coordinate is the multiple of difference change and the vertical coordinate is the significance of difference. The box shows the names of some genes with the highest significance level.

**Figure 5 plants-14-01736-f005:**
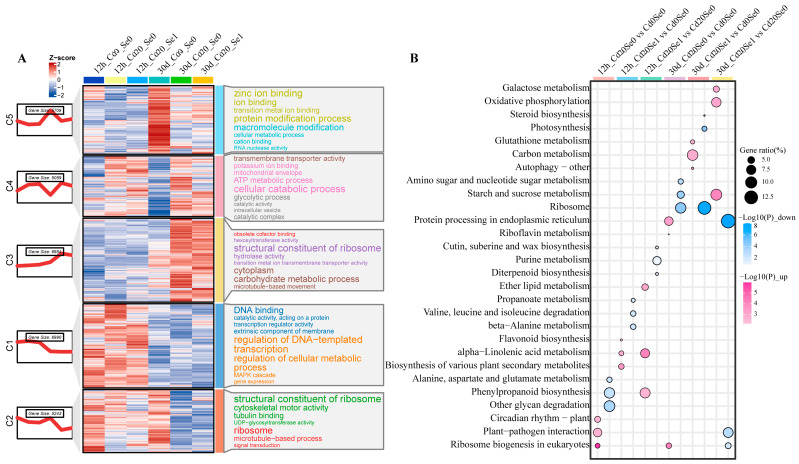
Trend analysis and enrichment annotation of DEGs. (**A**) C1–C5 gene clusters were identified by Mfuzz analysis and the first 8 gene ontology (GO) items of each gene cluster were annotated (*p* < 0.05). In these comments, the font size of each gene cluster is inversely proportional to its *p*-value, with the larger the font size, the smaller the *p*-value. (**B**) Enrichment bubble maps of the first 3 KEGG pathway items of different comparison gene clusters were obtained based on the Upset results (red bubbles represent upregulation; blue bubbles represent downregulation).

**Figure 6 plants-14-01736-f006:**
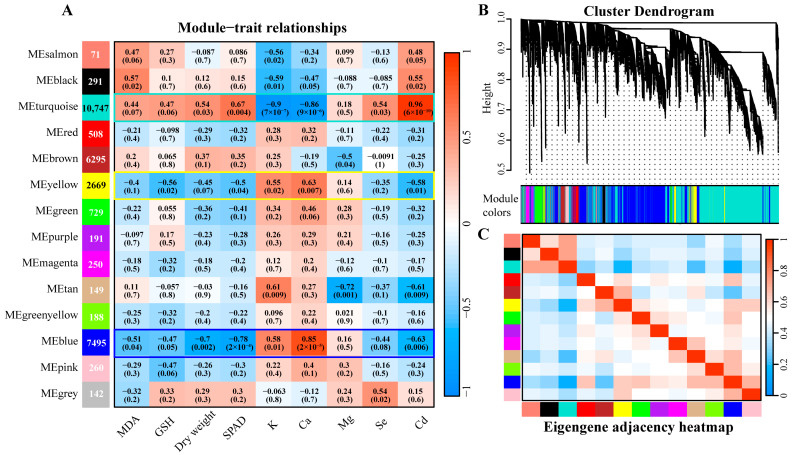
Weighted gene co-expression network analysis (WGCNA) was performed for DEGs using FPKM > 1. (**A**) The relationship between the co-expression module and physiological indicators: the number in the left module represents the number of DEGs in this module. The heat map shows the correlation between co-expression modules (vertical axis) and traits (horizontal axis), with a negative correlation in blue and a positive correlation in red. The *p*-value is shown in parentheses. (**B**) Hierarchical clustering tree of 14 co-expressed gene modules. (**C**) Heat maps of correlations between 14 modules, with negative correlations shown in blue and positive correlations shown in red.

**Figure 7 plants-14-01736-f007:**
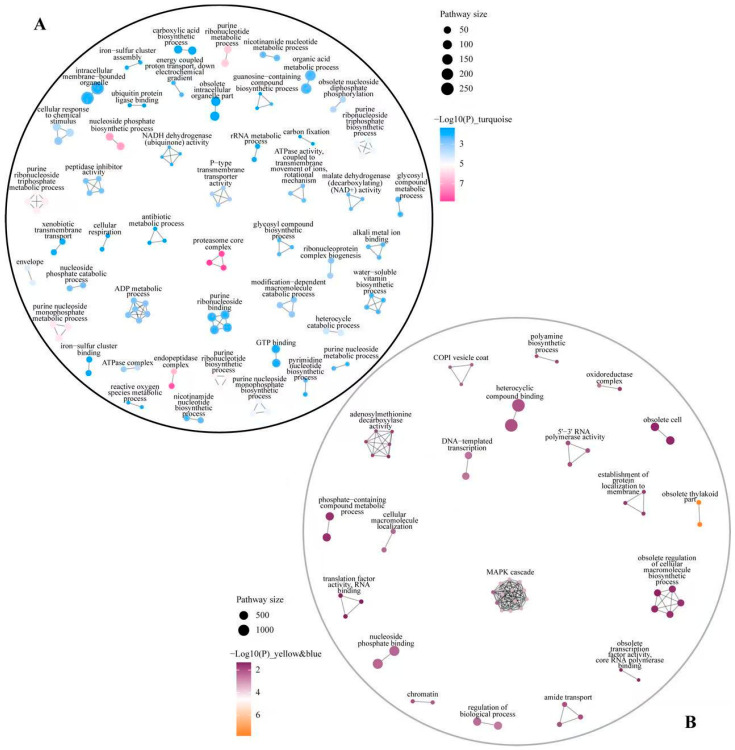
GO network analysis of DEGs in the turquoise module, integrated yellow module, and blue module. (**A**) turquoise module. (**B**) yellow + blue module. Note: Nodes represent significant paths and edges represent similarities between them.

**Figure 8 plants-14-01736-f008:**
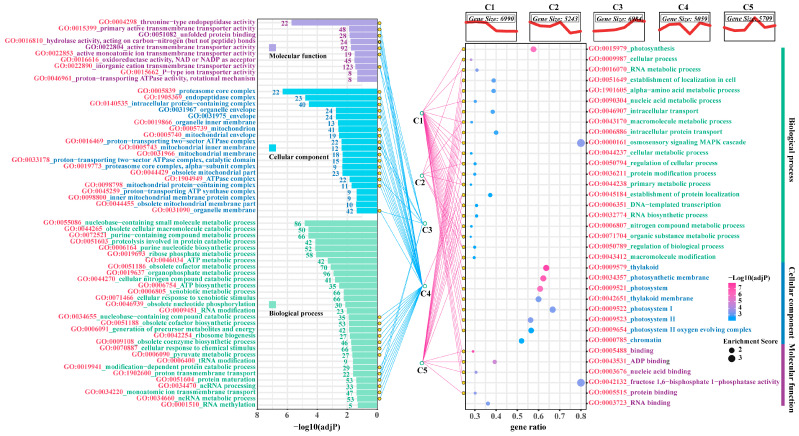
Significantly enriched GO items (*p* < 0.05) in the turquoise modules (**left**) and the combined yellow and blue modules (**right**). Note: The GO terms that are enriched in the turquoise module and the yellow + blue module are also found to be significantly enriched in the different trend gene sets (C1–C5) analyzed by Mfuzz.

**Figure 9 plants-14-01736-f009:**
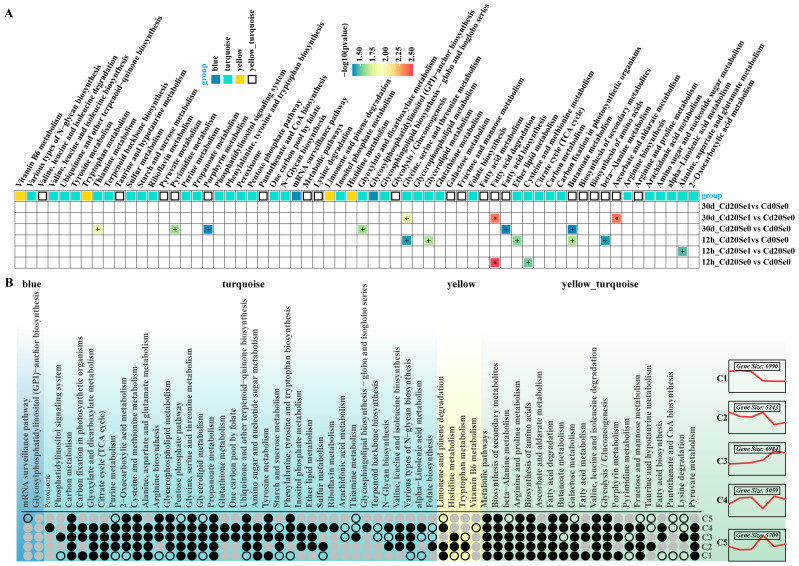
The KEGG enrichment results of the gene sets of the turquoise module and the yellow and blue module. (**A**) After KEGG enrichment analysis of the gene sets of the turquoise module and the yellow and blue module, the significant enrichment pathway (*p* < 0.05) was also significantly enriched in the gene sets of different comparison groups (+ *p* < 0.05, * *p* < 0.01). (**B**) After KEGG enrichment analysis of the gene sets of the turquoise module and the yellow and blue module, the significant enrichment pathway (*p* < 0.05) was also significantly enriched in different trend gene sets of Mfuzz.

**Figure 10 plants-14-01736-f010:**
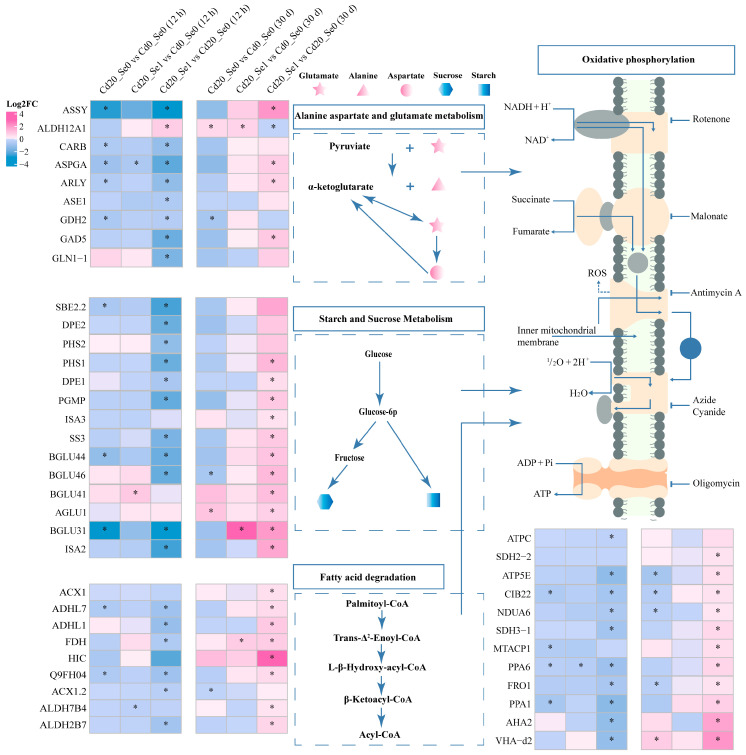
The expression of DEGs in the pathways of alanine, aspartate, and glutamate metabolism, starch and sucrose metabolism, fatty acid degradation, and oxidative phosphorylation in *G. uralensis* under different treatment conditions. Note: “*” indicates that the differential gene was significantly differentially expressed between the two treatments (*p* < 0.05).

**Figure 11 plants-14-01736-f011:**
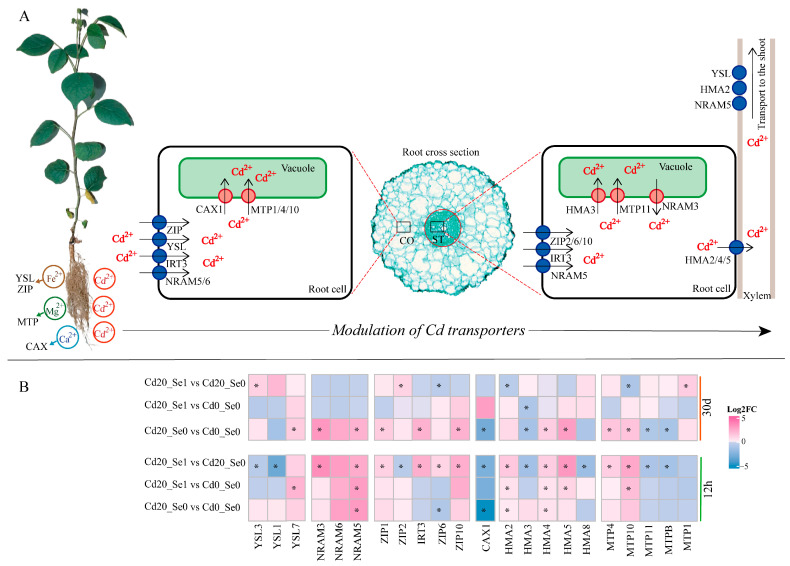
(**A**) Schematic diagram of the DEGs involved in Cd^2+^ transport under Se and Cd treatments. The area inside the red circle in the root cross-section represents the stele (ST), while the area outside represents the cortex (CO). (**B**) Expression of DEGs related to Cd^2+^ and other metal ion transport in *G. uralensis* root under different treatment conditions. Note: “*” indicates that the differential gene was significantly differentially expressed between the two treatments (*p* < 0.05).

**Figure 12 plants-14-01736-f012:**
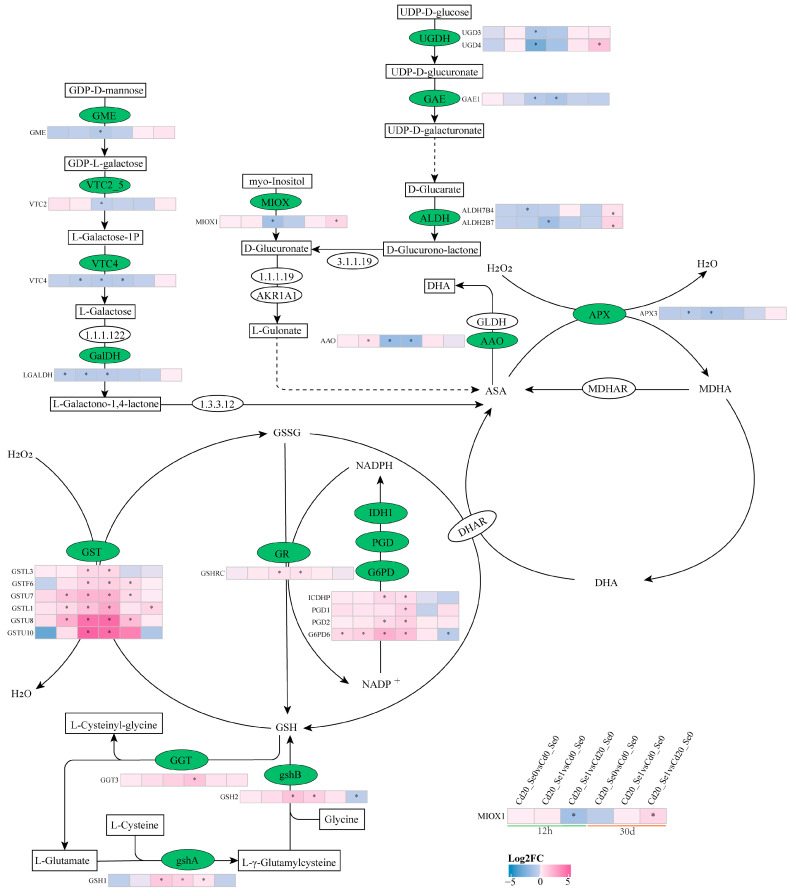
The expression of DEGs in the AsA-GSH pathways of *G. uralensis* under different treatment conditions. Dashed arrow: omitted intermediate steps. Note: “*” indicates that the differential gene was significantly differentially expressed between the two treatments (*p* < 0.05).

**Figure 13 plants-14-01736-f013:**
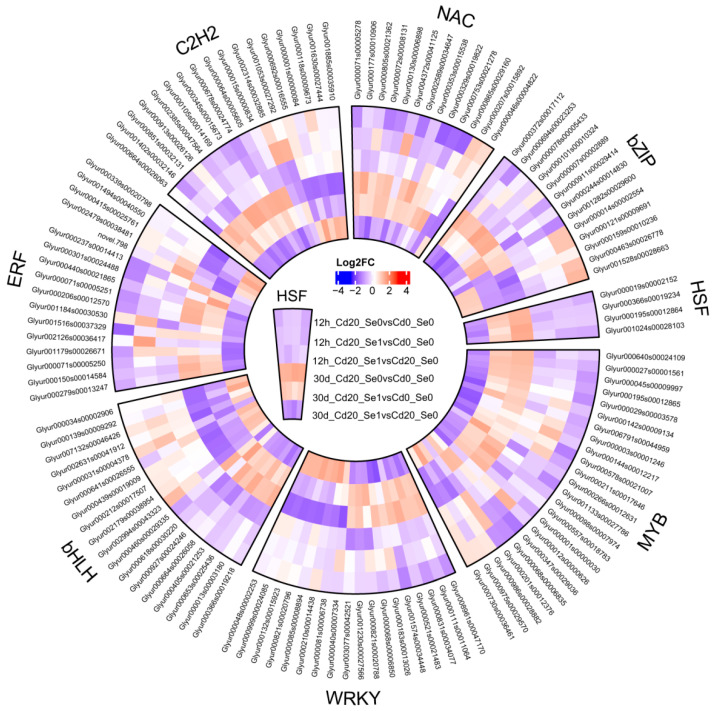
The expression of DEGs related to the transcription factors (MYB, WRKY, bHLH, ERF, C_2_H_2_, NAC, bZIP, and HSF) associated with Cd tolerance in the CK, Cd, and Cd + Se1 treatment groups.

**Table 1 plants-14-01736-t001:** The effects of different concentrations of selenium and cadmium on the root length of licorice.

	Length of Root (cm)	Cd0_Se0	Cd0_Se1	Cd0_Se5	Cd20_Se0	Cd20_Se1	Cd20_Se5
Cd0_Se0	13.61 ± 1.31	1					
Cd0_Se1	14.80 ± 2.19	0.075	1				
Cd0_Se5	11.78 ± 1.25	0.007 **	0.0001 **	1			
Cd20_Se0	9.22 ± 0.88	0.0001 **	0.0001 **	0.0001 **	1		
Cd20_Se1	10.92 ± 1.25	0.0001 **	0.0001 **	0.196	0.012 *	1	
Cd20_Se5	9.68 ± 1.05	0.0001 **	0.0001 **	0.002 **	0.489	0.063	1

Note: “*” indicates a difference between the two groups; the values of the root length are means ± SEs (*n*  =  3); * *p* < 0.05, ** *p* < 0.01.

**Table 2 plants-14-01736-t002:** The effects of different concentrations of selenium and cadmium on the SPAD value of licorice.

	Value of SPAD	Cd0_Se0	Cd0_Se1	Cd0_Se5	Cd20_Se0	Cd20_Se1	Cd20_Se5
Cd0_Se0	46.33 ± 2.22	1					
Cd0_Se1	50.52 ± 2.95	0.001 **	1				
Cd0_Se5	47.35 ± 3.17	0.4	0.011 *	1			
Cd20_Se0	37.90 ± 1.95	0.0001 **	0.0001 **	0.0001 **	1		
Cd20_Se1	45.20 ± 3.40	0.351	0.0001 **	0.079	0.0001 **	1	
Cd20_Se5	39.81 ± 2.05	0.0001 **	0.0001 **	0.0001 **	0.118	0.0001 **	1

Note: “*” indicates a difference between the two groups; SPAD values are means ± SEs (*n*  =  3); * *p* < 0.05, ** *p* < 0.01.

## Data Availability

Data are contained within the article.

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
