# Peer review of "Selenium Alleviates Cadmium Toxicity by Regulating Carbon Metabolism, AsA-GSH Cycle, and Cadmium Transport in Glycyrrhiza uralensis Fisch. Seedlings"

_plants, 2025, doi:10.3390/plants14121736_

Round 1
Reviewer 1 Report
Comments and Suggestions for Authors
The paper presents significant insights into how plants respond at the molecular level to cadmium (Cd) ion toxicity, as well as how selenium ions may help mitigate the effects of Cd-induced stress. However, I have a few comments regarding the content of this paper:
The clarity and resolution of Figures 4, 5, 7, 8, and 9 should be improved. At present, some graphical elements, labels, are difficult to read.
Section 2.3. In this section, the MDA and GSH contents were presented only for the 1 µM Se treatment, but not for the 5 µM dose. Could you clarify the reason for this omission? Including the data for the 5 µM Se treatment would be highly valuable, especially considering that this concentration may have different, possibly more adverse effects compared to the 1 µM dose, while still exhibiting positive effects relative to the control in terms of biomass, for example. Reporting these values would provide important insight into the dose-dependent responses and would be particularly relevant for optimizing selenium application in a practical context.
The information presented in Section 2.8 regarding the pathways and genes involved in cadmium ion transport is extremely valuable for understanding detoxification and homeostasis mechanisms. However, in the comparisons shown in Figures 10 and 11, it would have been useful to include a direct comparison between the Cd0_Se0 and Cd0_Se1 conditions. This comparison would have enabled a clearer identification of the genes and pathways specifically involved in the response to selenium (Se) in the absence of Cd, and more importantly, would have helped to detect potential interactions or increased predisposition of certain transporters for Se when Cd is also present. Such an approach would contribute to a more nuanced understanding of how Se may influence or modulate Cd ion transport at the molecular level.
Line 506: Should 'long-term' be in sentence case, or the point is this a mistake?
Line 555: Glycyrrhiza uralensis should be italicised.
I recommend italicizing the scientific names of all plant and microorganism species throughout the manuscript, including in the References section.
Author Response
请参阅附件。

Reviewer 2 Report
Comments and Suggestions for Authors
Dear Collegues
I was pleased to review the manuscript (Selenium alleviates cadmium toxicity by regulating carbon metabolism, AsA-GSH cycle, and cadmium transport in Glycyrrhiza uralensis Fisch. Seedlings). The topic of the study is actual, the subject is extremely interesting, environmental pollution by cadmium is a significant threat to society. The manuscript is well illustrated. It contains new experimental data.
The authors studied possible defense mechanisms of selenium on Glycyrrhiza uralensis plants under cadmium stress. The great advantage of this work is that the authors did not limit themselves to analyzing only molecular criteria of plant response to the treatments used, but also applied some physiological indicators. First of all, they showed that selenium reduces the inhibitory effect of Cd on growth processes. The authors obtained some experimental evidence that the protective effect of selenium may be based on (1) an increase in plant antioxidant activity due to the regulation of genes involved in the AsA-GSH cycle; (2) reduction of Cd content in roots, as evidenced by selenium regulation of genes involved in maintaining cellular homeostasis of cadmium (we are talking about cadmium transporters responsible for the uptake and release of cadmium from cells) and genes whose products are involved in sequestration of cadmium ions into vacuole.
I believe that the manuscript can be published after minor revisions and comments on some observations.
- The authors draw a significant part of their conclusions on the basis of transcriptome analysis, but, unfortunately, they did not validate the data obtained by this method, although it is mandatory.
- In the Discussion section there are the following subheadings: Se enhances carbon metabolism in G. uralensis seedlings under Cd stress; Se regulates the AsA-GSH cycle to reduce oxidative damage in G. uralensis; Se improves Cd tolerance in G. uralensis seedlings by regulating transport proteins. From my point of view, such formulations are not very correct, since the authors investigated only the genes involved in the above processes and not carbon metabolism, AsA-GSH cycle or transport proteins.
3 The authors determined the MDA content using thiobarbituric acid. Unfortunately, other compounds interact with thiobarbituric acid in addition to MDA due to its non-specificity, so it is common to use the term TBARS (tiobarbituric acid-reactive substances) instead of MDA. For this reason, I propose to replace MDA with TBARS.
- The authors treated the plants three times. I would like to know how many days later the successive treatments were carried out?
- How many plants were included in each experimental variant (each treatment group)?
- Centrifugation speed should be given in g, not rpm.
- 0.1 g of root tissue was used to determine GSH. From which zone of the root was the biological material taken?
Kind regards
Author Response
请参阅附件。
